# Associations between low Apgar scores and mortality by race in the United States: A cohort study of 6,809,653 infants

Emma Gillette [1,2]*, James P. Boardman [3,4], Clara Calvert [2,5], Jeeva John [4], Sarah J. Stock [2,3]

1 Arnhold Institute for Global Health at Icahn School of Medicine at Mount Sinai, New York, New York, United States of America, 2 University of Edinburgh Usher Institute, NINE Edinburgh BioQuarter, Edinburgh, United Kingdom, 3 MRC Centre for Reproductive Health, Queen's Medical Research Institute, Edinburgh, United Kingdom, 4 Centre for Clinical Brain Sciences, Chancellor's Building, Edinburgh, United Kingdom, 5 Department of Population Health, London School of Hygiene and Tropical Medicine, London, United Kingdom

* Emma.f.gillette@gmail.com

**Data Availability Statement:** Data for this manuscript was sourced from the CDC National Center for Health Statistics Vital Statistics Online Data Portal (https://www.cdc.gov/nchs/data_

## Abstract

### Background

Apgar scores measure newborn health and are strongly associated with infant outcomes, but their performance has largely been determined in primarily white populations. Given the majority of the global population is not white, we aim to assess whether the association between low Apgar score and mortality in infants varies across racial groups.

### Methods and findings

Population-based cohort study using 2016 to 2017 United States National Vital Statistics System data. The study included singleton infants born between $37^{+0}$ and $44^{+6}$ weeks to mothers over 15 years, without congenital abnormalities. We looked at 3 different mortality outcomes: (1) early neonatal mortality; (2) overall neonatal mortality; and (3) infant mortality. We used logistic regression to assess the association between Apgar score (categorized as low, intermediate, and normal) and each mortality outcome, and adjusted for gestational age, sex, maternal BMI, education, age, previous number of live births, and smoking status, and stratified these models by maternal race group (as self-reported on birth certificates). The cohort consisted of 6,809,653 infants (52.8% non-Hispanic white, 23.7% Hispanic, 13.8% non-Hispanic black, 6.6% non-Hispanic Asian, and 3.1% non-Hispanic other). A total of 6,728,829 (98.8%) infants had normal scores, 63,467 (0.9%) had intermediate scores, and 17,357 (0.3%) had low Apgar scores. Compared to infants with normal scores, low-scoring infants had increased odds of infant mortality. There was strong evidence that this association varied by race ($p < 0.001$) with adjusted odds ratios (AORs) of 54.4 (95% confidence interval [CI] 49.9 to 59.4) in non-Hispanic white, 70.02 (95% CI 60.8 to 80.7) in Hispanic, 23.3 (95% CI 20.3 to 26.8) in non-Hispanic black, 100.4 (95% CI 74.5 to 135.4) in non-Hispanic Asian, and 26.8 (95% CI 19.8 to 36.3) in non-Hispanic other infants. The main limitation was missing data for some variables, due to using routinely collected data.

access/vitalstatsonline.htm). All data and
associated data dictionaries are accessible online.

**Funding:** SJS is funded by a Wellcome Trust
Clinical Career Development Fellowship 209560/Z/
17/Z (https://wellcome.org). The funders had no
role in study design, data collection and analysis,
decision to publish, or preparation of the
manuscript.

**Competing interests:** I have read the journal's
policy and the authors of this manuscript have the
following competing interests:SJS has received
grant funding, paid to her institution, from the
Wellcome Trust, The National Institute of
Healthcare Research, The Chief Scientist Office
Scotland and Tommy's Charity. SJS is an academic
Editor for PLOS Medicine.

**Abbreviations:** AOR, adjusted odds ratios; BMI,
body mass index; CDC, Centers for Disease Control
and Prevention; CI, confidence interval; GED,
General Educational Development; NCHS, National
Center for Health Statistics; OR, odds ratio.

## Conclusions

The association between Apgar scores and mortality varies across racial groups. Low Apgar scores are associated with mortality across racial groups captured by United States (US) records, but are worse at discriminating infants at risk of mortality for black and non-Hispanic non-Asian infants than for white infants. Apgar scores are useful clinical indicators and epidemiological tools; caution is required regarding racial differences in their applicability.

## Author summary

### Why was this study done?

- Apgar scores are commonly used indicators of infants' well-being at birth and as predictors of mortality and long-term disability.

- Apgar scores have been validated in predominantly white populations.

- The impact of race on the relationship between Apgar score and early neonatal (death within 0 to 6 days of birth), overall neonatal (death within 0 to 27 days), and infant mortality (death within 1 year) is unknown.

### What did the researchers do and find?

- We conducted a cohort study of all singleton, term-birth infants born to mothers over 15 years old, without congenital abnormalities, in the United States (US) in 2017 and 2018.

- Our analyses illustrated that race was associated with the assignment of the Apgar score category and with all categories (early neonatal, overall neonatal, and infant mortality) of mortality.

- There was a stronger association between low Apgar score and all categories of mortality among non-Hispanic white, non-Hispanic Asian and Hispanic infants than in non-Hispanic black and non-Hispanic other infants.

### What do these findings mean?

- These findings suggest that Apgar scores are less useful for estimating the odds of mortality for non-Hispanic black and non-Hispanic non-Asian infants than for non-Hispanic white, non-Hispanic other infants.

- Apgar scores are useful predictors of morbidity and mortality; however, their association with mortality is influenced by infant race. Further work to understand which components of the score explain differential associations is needed for developing a scoring system that performs equally well across racial groups.

## Introduction

The Apgar score has been used for nearly 70 years to measure infant health and physical well-being immediately after birth and as a predictor of mortality and indicator of an infant's response to resuscitative efforts [1,2]. Apgar scores are widely used in epidemiological studies for providing population-level information about infants' status at birth, predicting neurodevelopmental outcomes and infant mortality and as surrogate markers of morbidity [3–5]. In these contexts, Apgar scores are applied across populations, but the tool was developed and validated in predominantly white populations.

The score is composed of 5 variables, each with a value of 0, 1, or 2 [6]. The variables are: heart rate, respiratory effort, muscle tone, reflex response, and skin coloring, each assessed at 1, 5 and 10 minutes after birth [6]. Of these, the 5-minute score is regarded as the best predictor of infant mortality and is commonly used in epidemiological studies and trials [6,7]. The overall Apgar score ranges from 0 to 10 and is frequently categorized as low (0 to 3), intermediate (4 to 6), or normal (7 to 10) [6]. These categorizations of the score were originally suggested by Dr. Virginia Apgar and allow comparison between similar studies [4,7,8].

Few studies validating the use of Apgar scores to assess infants' status at birth or to look at the association between Apgar scores and adverse infant outcomes have considered race and ethnicity and none have specifically considered white, black, Asian, Hispanic, and other racial groups individually. Two studies have reported that black infants were assigned lower Apgar scores than white infants [9,10]. Some studies have speculated that this may be due to differential interpretation of the skin color variable in infants that are not of white ethnicity [8–10]. Additionally, a study reported that 1-minute Apgar scores were the strongest predictors of infant mortality for Mexican American infants, the worst for black infants, with an intermediate ability for white infants [8].

The lack of understanding around the application of Apgar scores across different race groups is surprising given the wide racial disparities that exist in birth outcomes across many settings including the United States (US) [8,11–15]. Black infants have an infant mortality rate of more than twice that of white infants in the US and have consistently been found to have higher incidences of adverse birth outcomes such as preterm birth, low birth weight, and being small for gestational age [11,16]. Drivers of poor pregnancy outcomes in some race groups are complex and are likely to reflect the interplay of multiple impacts of structural racism including socioeconomic inequalities, access to quality healthcare, and discrimination [17–19].

In light of the widespread use of Apgar scores in clinical and epidemiological settings, the lack of research on racial differences in the applicability of the scores and the prevalence of racial disparities in birth outcomes in the US and elsewhere, this study aims to report on the associations between maternal race, 5-minute Apgar scores, and infant mortality. The objectives of this study are to evaluate the association between maternal race and 5-minute Apgar score, the association between maternal race and mortality, and whether there is a differential association between 5-minute Apgar scores and mortality by race. We hypothesize that the associations between 5-minute Apgar scores and early neonatal, overall neonatal, and infant mortality differ by race.

## Methods

### Study design and setting

This population cohort study evaluated all infants born between January 1, 2016 and December 31, 2017 in the US ($n = 7,820,866$). This study is reported as per the Strengthening the Reporting of Observational Studies in Epidemiology (STROBE) guidelines (S1 Table).

## Study participants

Inclusion criteria were single births of infants between $37^{+0}$ and $44^{+6}$ weeks to mothers older than 15 years who were residents of the US Births were excluded if they had no recorded gestational age, 5-minute Apgar score, or maternal race. Missing values for maternal age were imputed by National Center for Health Statistics (NCHS) as the age of the mother from the previous birth record of the same race and birth order in 0.01% of births, and missing values for plurality were imputed by NCHS as singleton births in 0.004% of births. As congenital abnormalities can affect birth outcomes, infants were excluded if they were born with any major congenital abnormality, which were identified by NCHS to include: anencephaly, meningomyelocele/spina bifida, cyanotic congenital heart disease, congenital diaphragmatic hernia, omphalocele, gastroschisis, limb reduction defect, cleft palate, Down syndrome, suspected chromosomal disorder, and hypospadias. This analysis was restricted to term births because elements of the Apgar score, including tone, color, and reflex irritability are dependent on physiological maturity, and recommendations on its use in preterm populations vary [20].

## Data sources

All data were nonidentifiable and publicly accessible through the NCHS Division of Vital Statistics cohort-linked birth and death database, and complied with the NCHS, Centers for Disease Control and Prevention (CDC) Data User Agreement Terms and Conditions [21–23]. The database is composed of data collected directly from US Standard Birth and Death Certificates, including demographic information, and is commonly used in CDC Reports and national studies on infant mortality and neonatal outcomes [24,25]. NCHS linked birth certificate and death certificate data using linking identification numbers, resulting in 22,197 (99.6%) linked death records.

## Variables

The outcome of interest was mortality across the first year of life, subdivided into early neonatal mortality (death within 0 to 6 days of birth), overall neonatal mortality (death within 0 to 27 days of birth), and infant mortality (death within 1 year of birth).

The explanatory variables were 5-minute Apgar score and maternal race/ethnicity, as a surrogate for infant race due to a high frequency of missing data for paternal race. Apgar score was measured by a birth attendant 5 minutes after birth, recorded on the infant's medical chart, and transcribed to the birth certificate by hospital staff using an NCHS facility worksheet [23]. The same worksheet was completed for births outside the hospital [23]. Scores were recorded as whole numbers in the birth certificates and were categorized for this analysis as low (0 to 3), intermediate (4 to 6) and normal (7 to 10). Maternal race was self-reported by the mother by choosing 1 or more of 15 race categories and 5 Hispanic origin categories from the NCHS facility worksheet [23]. The combined race/ethnicity variable used in this analysis was categorized based on the NCHS designations in order to analyze racial groups that can be compared across studies (S2 Table). This analysis considered Hispanic as a separate race/ethnicity category in accordance with NCHS guidelines for this dataset, which utilize single-race categorizations.

All multivariable regression models adjusted for the confounding effects of the following covariates: gestational age (continuous scale of whole numbered weeks), fetal sex (male or female), maternal educational attainment (≤eighth grade, ninth to 12th grade without diploma, High School/ General Educational Development (GED), Associate's degree, Bachelor's degree, Master's degree, Doctorate or Professional degree, unknown), maternal body mass index (BMI) (underweight [$<18.5$ kg/m$^2$], normal [18.5 to 24.9 kg/m$^2$], overweight [25 to 29.9 kg/m$^2$], obesity I [30 to 34.9 kg/m$^2$], obesity II [35 to 39.9 kg/m$^2$], extreme obesity III

[$\geq$40 kg/m$^2$], unknown), maternal smoking status (smoker or nonsmoker), maternal age (15 to 19, 20 to 24, 25 to 29, 30 to 34, 35 to 39, and 40+), and number of previous live births (0, 1 to 2, 3 to 4, and 5+). Missing values for maternal education, birth weight, smoking status, previous number of live births, and maternal BMI were included as "unknown" categories.

## Statistical methods

The data were analyzed using R version 4.0.2. No overall prospective analysis plan was used. Demographic characteristics were derived for the cohort and each Apgar score group (low, intermediate, normal), with frequencies and percentages reported for categorical variables and means and standard deviations reported for continuous variables.

Univariate logistic regression models were used to quantify the association between each racial group and odds of being assigned a low Apgar score (low versus not low), being assigned an intermediate Apgar score (intermediate versus not intermediate) and to quantify the association between race group and each mortality outcome (early neonatal mortality, neonatal mortality, and infant mortality). Univariate logistic regression models were also used to assess the crude association between each covariate and the mortality outcomes, stratified by race group.

Multivariable logistic regression models were conducted to determine the association between Apgar score and each mortality outcome in the total population and stratified by race group. We formally assessed whether there was evidence that the association between Apgar score and mortality varied by race group by including an interaction term in the adjusted model in the total population. Additionally, we conducted a chi-squared test to determine whether there were trends between Apgar score category and early neonatal, overall neonatal, and infant mortality rates among each race group.

## Ethics committee approval

The study was sponsored by the University of Edinburgh (reference AC20095). Prior to commencement, the research was subject to the Usher Institute (University of Edinburgh) ethics and data protection oversight process. The ethics and data protection triage and overview self-audit of ethics and data protection issues (completed by EG) confirmed that the proposed research, being secondary analysis of a fully anonymized publicly accessible dataset, posed no foreseeable ethics or data protection risks. This indicated there was no requirement for proceeding to a full formal ethics and data protection review by the Usher Research Ethics Group.

## Results

### Descriptive cohort characteristics

The NCHS cohort-linked database recorded 7,820,866 live births between January 1, 2016 and December 31, 2017. As shown in Fig 1, 6,809,653 births were eligible for inclusion in this study. Data were missing for Apgar score (0.4%), maternal race (0.9%), and gestational age (0.8%), and these cases were excluded from analysis.

A total of 17,357 (0.3%) newborns had low Apgar scores, 63,467 (0.9%) had intermediate scores, and 6,728,829 (98.8%) had normal scores (Table 1). There were 2,115 (0.03%) early neonatal deaths, 3,811 (0.06%) overall neonatal deaths, and 12,436 (0.2%) infant deaths.

Table 1 describes the cohort characteristics. Overall, 49.1% infants were female, 52.8% were non-Hispanic white, 23.7% were Hispanic, 13.8% were non-Hispanic black, 6.6% were non-Hispanic Asian, 3.1% were non-Hispanic other, and 78.2% were born to mothers aged 20 to 34. The majority of mothers (86.0%) had a high school degree or higher, and the average gestational age was 39 weeks. Data were missing and included as an "unknown" category for

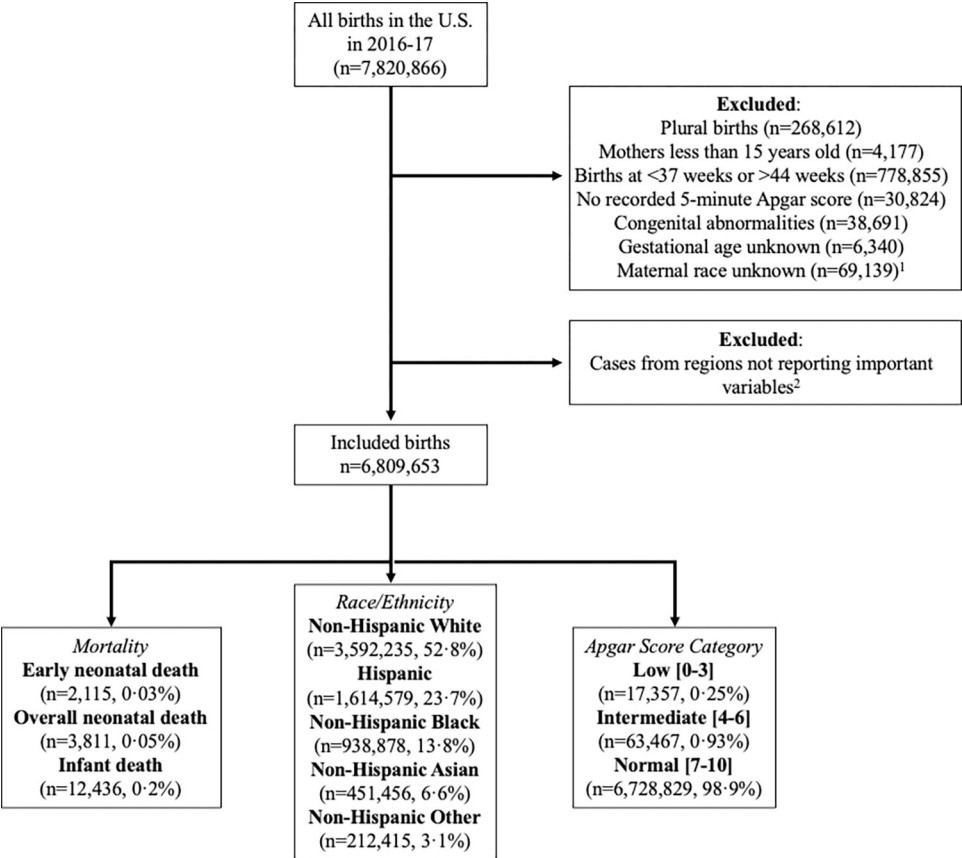

**Fig 1. Flowchart of exclusions from study population to analysis cohort.** [1]Exclusion criteria overlapped, these values represent the frequency of cases in the study population, not necessarily the frequency of excluded cases. [2]Important variables included maternal race, maternal education, number of prenatal care visits, smoking status, and 5-minute Apgar score.

maternal education (0.8%), birth weight (0.03%), smoking status (0.4%), previous live births (0.3%), and maternal BMI (2.4%).

## Association between maternal race and 5-minute Apgar score

Race group was associated with the assignment of Apgar score category ($p < 0.001$). Compared with non-Hispanic white infants, non-Hispanic black infants had 1.7 times the odds of being assigned a low Apgar score (95% confidence interval [CI] 1.6 to 1.8) and non-Hispanic other infants had 1.3 times the odds (95% CI 1.2 to 1.4) (Table 2). Non-Hispanic Asian and Hispanic infants had 23% (95% CI 0.7 to 0.8) and 30% (95% CI 0.6 to 0.8) lower odds of being assigned low scores than non-Hispanic white infants, respectively.

## Association between maternal race and mortality

There was strong evidence of an association between maternal race and mortality across the first year of life. Compared with non-Hispanic white infants, non-Hispanic black infants had higher odds of all categories of mortality (early neonatal mortality: odds ratio [OR] 1.3 [95% CI 1.2 to 1.5]; overall neonatal mortality: OR 1.5 [95% CI = 1.3 to 1.6]; infant mortality OR 1.9 [95% CI = 1.8 to 2.0]; S4 Table). Conversely, non-Hispanic Asian and Hispanic infants had lower odds for all mortality outcomes when compared with white infants. There was no

**Table 1. Descriptive characteristics of study cohort, stratified by 5-minute Apgar score.**

| | | Total Cohort | Low (0–3) | Intermediate (4–6) | Normal (7–10) | Imputed/ Missing |
|---|---|---|---|---|---|---|
| | | *n* = 6,809,653 | *n* = 17,357 | *n* = 63,467 | *n* = 6,728,829 | |
| Year of birth | | | | | | |
| | 2016 | 3,443,416 (50·6%) | 8,825 (50·8%) | 32,038 (50·5%) | 3,402,553 (50·6%) | - |
| | 2017 | 3,366,237 (49·4%) | 8,532 (49·2%) | 31,429 (49·5%) | 3,326,276 (49·4%) | |
| Sex | | | | | | |
| | Male | 3,466,945 (50·9%) | 9,959 (57·4%) | 35,666 (56·2%) | 3,421,320 (50·8%) | 33[1] |
| | Female | 3,342,708 (49·1%) | 7,398 (42·6%) | 27,801 (43·8%) | 3,307,509 (49·2%) | |
| Maternal race | | | | | | |
| | Non-Hispanic white | 3,592,235 (52·8%) | 8,863 (51·1%) | 36,144 (56·9%) | 3,547,228 (52·7%) | - |
| | Hispanic | 1,614,579 (23·7%) | 3,080 (17·7%) | 10,124 (16·0%) | 1,601,375 (23·8%) | |
| | Non-Hispanic black | 938,878 (13·8%) | 3,931 (22·6%) | 11,816 (18·6%) | 923,131 (13·7%) | |
| | Non-Hispanic Asian | 451,546 (6·6%) | 785 (4·5%) | 2,919 (4·6%) | 447,842 (6·7%) | |
| | Non-Hispanic other | 212,415 (3·1%) | 698 (4·2%) | 2,464 (3·9%) | 209,253 (3·1%) | |
| Maternal age | | | | | | |
| | 15–19 | 355,849 (5·2%) | 1,370 (7·9%) | 4,633 (7·3%) | 349,846 (5·2%) | 152[1] |
| | 20–24 | 1,387,621 (20·4%) | 4,119 (23·7%) | 14,765 (23·3%) | 1,368,737 (20·3%) | |
| | 25–29 | 2,007,175 (29·5%) | 4,885 (28·1%) | 18,235 (28·7%) | 1,984,055 (29·5%) | |
| | 30–34 | 1,924,013 (28·3%) | 4,226 (24·3%) | 15,911 (25·1%) | 1,903,876 (28·3%) | |
| | 35–39 | 936,203 (13·7%) | 2,225 (12·8%) | 7,978 (12·6%) | 926,000 (13·8%) | |
| | >40 | 198,792 (2·9%) | 532 (3·1%) | 1,945 (3·1%) | 196,315 (2·9%) | |
| Maternal education | | | | | | |
| | <Eighth grade | 223,993 (3·3%) | 576 (3·3%) | 1,851 (2·9%) | 221,566 (3·3%) | - |
| | Ninth–12th grade, no diploma | 679,548 (10·0%) | 1,890 (10·9%) | 6,585 (10·4%) | 671,073 (10·0%) | |
| | High school/GED | 1,704,066 (25·0%) | 4,818 (27·8%) | 16,764 (26·4%) | 1,682,484 (25·0%) | |
| | Some college credit | 1,389,787 (20·4%) | 3,863 (22·3%) | 14,099 (22·2%) | 1,371,825 (20·4%) | |
| | Associate's | 559,879 (9·2%) | 1,380 (8·0%) | 5,494 (8·7%) | 553,005 (8·2%) | |
| | Bachelor's | 1,386,594 (20·4%) | 3,038 (17·5%) | 11,742 (18·5%) | 1,371,814 (20·4%) | |
| | Master's | 630,970 (9·3%) | 1,241 (7·1%) | 4,972 (7·8%) | 624,757 (9·3%) | |
| | Doctorate/professional | 180,718 (2·7%) | 358 (2·1%) | 1,468 (2·3%) | 178,892 (2·7%) | |
| | Unknown | 54,098 (0·8%) | 193 (1·1%) | 4,920 (0·8%) | 53,413 (0·8%) | |
| Infant birth weight (g) | | | | | | |
| | <1500 | 1,184 (0·02%) | 47 (0·3%) | 82 (0·1%) | 1,055 (0·02%) | - |
| | 1,500–1,999 | 9,023 (0·1%) | 171 (1·0%) | 353 (0·6%) | 8,499 (0·1%) | |
| | 2,000–2,499 | 156,531 (2·3%) | 857 (4·9%) | 2,545 (4·0%) | 153,129 (2·3%) | |
| | 2,500–2,999 | 1,160,490 (17·0%) | 3,462 (19·9%) | 11,701 (18·4%) | 1,145,327 (17·0%) | |
| | 3,000–3,499 | 2,866,785 (41·1%) | 6,343 (36·5%) | 24,047 (37·9%) | 2,836,395 (42·2%) | |
| | 3,500–3,999 | 2,014,972 (29·6%) | 4,580 (26·4%) | 17,663 (27·8%) | 1,992,729 (29·6%) | |
| | 4,000–4,499 | 517,583 (7·6%) | 1,409 (8·1%) | 5,620 (8·9%) | 510,554 (7·6%) | |
| | 4,500–4,999 | 72,522 (1·1%) | 370 (2·1%) | 1,159 (1·8%) | 70,993 (1·1%) | |
| | ≥5,000 | 8,302 (0·1%) | 77 (0·4%) | 237 (0·4%) | 7,988 (0·1%) | |
| | Unknown | 2,261 (0·03%) | 41 (0·2%) | 60 (0·1%) | 2,160 (0·03%) | |
| Smoking status | | | | | | |
| | No smoking | 6,320,204 (92·8%) | 15,670 (90·3%) | 57,312 (90·3%) | 6,247,222 (92·8%) | - |
| | Smoking | 460,417 (6·8%) | 1,566 (9·0%) | 5,758 (9·1%) | 453,093 (6·7%) | |
| | Unknown | 29032 (0·4%) | 121 (0·7%) | 397 (0·6%) | 28,514 (0·4%) | |
| Maternal BMI | | | | | | |

*(Continued)*

**Table 1.** (Continued)

| | | Total Cohort | Low (0–3) | Intermediate (4–6) | Normal (7–10) | Imputed/ Missing |
|---|---|---|---|---|---|---|
| | Underweight (<18·5) | 225,679 (3·3%) | 495 (2·9%) | 1,786 (2·8%) | 223,398 (3·3%) | - |
| | Normal (18·5–24·9) | 2,942,814 (43·2%) | 6,486 (37·4%) | 24,716 (38·9%) | 2,911,612 (43·3%) | |
| | Overweight (25–29·9) | 1,744,211 (25·6%) | 4,309 (24·8%) | 15,948 (25·1%) | 1,723,954 (25·6%) | |
| | Obesity I (30–34·9) | 954,107 (14·0%) | 2,685 (15·5%) | 9,574 (15·1%) | 941,848 (14·0%) | |
| | Obesity II (35–39·9) | 459,520 (6·7%) | 1,465 (8·4%) | 5,284 (8·3%) | 452,771 (6·7%) | |
| | Obesity III (≥40) | 319,681 (4·7%) | 1,342 (7·7%) | 4,563 (7·2%) | 313,776 (4·7%) | |
| | Unknown | 163,641 (2·4%) | 575 (3·3%) | 1,596 (2·5%) | 161,470 (2·4%) | |
| Previous live births | | | | | | |
| | 0 | 2,611,902 (38·4%) | 9,708 (55·9%) | 35,295 (55·6%) | 2,566,899 (38·1%) | |
| | 1–2 | 3,359,993 (49·3%) | 5,751 (33·1%) | 21,667 (34·1%) | 3,332,575 (49·5%) | |
| | 3–4 | 672,440 (9·9%) | 1,378 (7·9%) | 4,948 (7·8%) | 666,114 (9·9%) | |
| | 5+ | 146,832 (2·2%) | 421 (2·4%) | 1,293 (2·0%) | 145,118 (2·2%) | |
| | Unknown | 18,486 (0·3%) | 99 (0·6%) | 264 (0·4%) | 18,123 (0·3%) | |
| Gestational age | | | | | | |
| | Mean (SD) | 39·0 (1·1) | 39·0 (1·2) | 39·0 (1·2) | 39·0 (1·1) | - |
| Mortality category | | | | | | |
| Early neonatal death (0–6 days) | N (Rate per 1,000 births) | 2,115 (0·3) | 1,029 (59·3) | 362 (5·7) | 724 (0·1) | - |
| Neonatal death (0–27 days) | N (Rate per 1,000 births) | 3,811 (0·6) | 1,190 (68·6) | 546 (8·6) | 2,075 (0·3) | - |
| Infant death (0–365 days) | N (Rate per 1,000 births) | 12,436 (1·8) | 1,333 (76·8) | 823 (13·0) | 10,280 (1·5) | - |

[1]Missing values imputed by NCHS.

BMI, body mass index; GED, General Educational Development; SD, standard deviation.

evidence of a difference in early neonatal mortality between non-Hispanic other and white infants, but non-Hispanic other infants had higher odds of neonatal and infant mortality.

## Impact of race on the relationship between 5-minute Apgar score and mortality

The early neonatal, overall neonatal, and infant mortality rates decreased with increasing Apgar scores in all races (S3 Table). Across all races, low Apgar score was a strong risk factor

**Table 2. Unadjusted odds ratio of being assigned a low (0–3) or intermediate (4–6) Apgar score by maternal race.**

| Race group | Assigned low score | Odds ratio of low Apgar score (95% CI) | P value* | Assigned intermediate score | Odds ratio of intermediate score (95% CI) | P value* |
|---|---|---|---|---|---|---|
| Non-Hispanic white (n = 3,592,235) | 8,863 (0·3%) | 1 (ref) | - | 36,144 (1·01%) | 1 (ref) | - |
| Hispanic (n = 1,614,579) | 3,080 (0·2%) | 0·77 (0·74–0·81) | <0.001 | 10,124 (0·6%) | 0·62 (0·61–0·63) | <0.001 |
| Non-Hispanic black (n = 938,878) | 3,931 (0·4%) | 1·70 (1·64–1·76) | <0.001 | 11,816 (1·3%) | 1·25 (1·23–1·28) | <0.001 |
| Non-Hispanic Asian (n = 451,546) | 785 (0·2%) | 0·70 (0·65–0·76) | <0.001 | 2,919 (0·7%) | 0·64 (0·62–0·66) | <0.001 |
| Non-Hispanic other (n = 212,415) | 698 (0·3%) | 1·33 (1·23–1·44) | <0.001 | 2,464 (1·2%) | 1·15 (1·11–1·20) | <0.001 |

*Wald p-value

CI, confidence interval.

for mortality across the first year of life. Low Apgar score was a stronger risk factor than an intermediate score, and there was a strong association between score category and all categories of mortality ($p < 0.001$; S3 Table).

There was strong evidence that the adjusted association between low Apgar score and mortality varied by race group ($p < 0.001$) for all mortality outcomes. The adjusted odds ratio (AOR), comparing the odds of infant mortality among infants with low Apgar scores to those with normal Apgar scores, were higher in non-Hispanic Asian (AOR 100.4 [95% CI 74.5 to 135.4]), non-Hispanic white (AOR 54.4 [95% CI 49.9 to 59.4]), and Hispanic (AOR 70.0 [95% CI 60.8 to 80.7]) groups than non-Hispanic black (AOR 23.3 [95% CI 20.3 to 26.8]) and non-Hispanic other (AOR 26.8 [95% CI 19.8 to 36.3]) groups (Table 3). Similar associations were present in the early neonatal and overall neonatal mortality categories, with non-Hispanic black and non-Hispanic other groups consistently having the lowest odds ratio for the association between low Apgar score and mortality (Fig 2).

Similar trends of lesser magnitudes persisted in the associations between intermediate scores and all categories of mortality (Table 3). The AORs for the associations between intermediate Apgar score and neonatal mortality varied across race groups ($p < 0.001$) with AORs of 22.0 (95% CI 19.2 to 25.3) for non-Hispanic white infants, 20.2 (95% CI 16.4 to 24.9) for non-Hispanic black infants, 32.5 (95% CI 20.2 to 52.5) for non-Hispanic Asian infants, 14.0 (95% CI 8.1 to 24.1) for non-Hispanic other infants, and 33.2 (95% CI 26.7 to 41.4) for Hispanic infants ($p$-value $< 0.001$). The differences between the race groups in these associations increased across the first year of life (Table 3). The final adjusted models, stratified by race group, are presented in S5–S9 Tables.

## Discussion

Overall, we find that low and intermediate Apgar scores are strongly associated with mortality across the first year of life in the US. These findings align with Dr. Apgar's original use of the score to predict neonatal mortality and support the use of the 5-minute Apgar score in research [4,7,26]. These data also illustrate for the first time, to our knowledge, that these strong associations persist across racial groups captured by US birth records. We do, however, also find evidence that there is variation in the relationship between Apgar score and mortality across different racial groups. Non-Hispanic black and non-Hispanic other infants have higher odds of mortality across the first year of life, and are more likely to be assigned a low Apgar score, when compared with white infants; yet, multivariable regression models revealed that the association between low Apgar score and mortality is weakest in non-Hispanic black and non-Hispanic other groups. These findings indicate that low Apgar scores are differentially associated with mortality across race groups and, more specifically, suggest that low Apgar scores are less good at discriminating the risk of mortality in non-Hispanic black and non-Hispanic other infants compared to other race groups.

The findings are consistent with literature suggesting that Apgar scores are strongly associated with mortality across the first year of life [6,13,27,28]. The results also align with research demonstrating that Apgar scores were more predictive of mortality in white and Mexican-American infants than in black infants [8]. This study expanded on the findings of previous research by evaluating more racial groups in a larger and more representative study population. These results add to a body of literature that suggests that the performance of mortality prediction tools in neonatal groups can be influenced by race, as demonstrated by the inclusion of race in the use of an estimator tool for bronchopulmonary dysplasia in preterm infants [29]. The mortality rates in our study population were lower than nationally reported estimates for 2018 of 3.75 neonatal deaths per 1,000 births and a rate of 5.64 infant deaths per 1,000

**Table 3. AORs for early neonatal mortality, neonatal mortality, and infant mortality in relation to Apgar score category, stratified by maternal race group.**

| | Early neonatal mortality (<7 days) | | | Overall neonatal mortality (<28 days) | | | Infant mortality (≤1 year) | | |
|---|---|---|---|---|---|---|---|---|---|
| | Deaths (rate per 1,000 births) | AOR (95% CI) | p-value* | Deaths (rate per 1,000 births) | AOR (95% CI) | p-value* | Deaths (rate per 1,000 births) | AOR (95% CI) | p-value* |
| **Overall** | | | | | | | | | |
| Normal (n = 6,728,829) | 724 (0·1) | 1 (ref) | | 2,075 (0·3) | 1 (ref) | | 10,280 (1·5) | 1 (ref) | |
| Intermediate (n = 63,467) | 362 (5·7) | 45·9 (40·3–52·2) | <0.001 | 546 (8·6) | 23·8 (21·6–26·2) | <0.001 | 823 (13·0) | 7·6 (7·02–8·1) | <0.001 |
| Low (n = 17,357) | 1,029 (59·3) | 493·1 (445·9–545·4) | <0.001 | 1,190 (68·6) | 199·4 (184·6–215·3) | <0.001 | 1,333 (76·8) | 47·5 (44·6–50·5) | <0.001 |
| **Non-Hispanic white** | | | | | | | | | |
| Normal (n = 3,547,228) | 349 (0·1) | 1 (ref) | | 1,047 (0·3) | 1 (ref) | | 4,952 (1·4) | 1 (ref) | |
| Intermediate (n = 36,144) | 183 (5·1) | 44·4 (37·0–53·4) | <0.001 | 278 (7·7) | 22·04 (19·2–25·3) | <0.001 | 396 (11·0) | 6·9 (6·2–7·6) | <0.001 |
| Low (n = 8,863) | 564 (63·6) | 598·9 (519·9–690·0) | <0.001 | 642 (72·4) | 225·0 (202·4–250·2) | <0.001 | 703 (79·3) | 54·4 (49·9–59·4) | <0.001 |
| **Hispanic** | | | | | | | | | |
| Normal (n = 1,601,375) | 172 (0·1) | 1 (ref) | | 431 (0·3) | 1 (ref) | | 1,886 (1·2) | 1 (ref) | |
| Intermediate (n = 10,124) | 74 (7·3) | 59·0 (44·5–78·2) | <0.001 | 107 (10·6) | 33·2 (26·7–41·4) | <0.001 | 161 (15·9) | 12·4 (10·5–14·7) | <0.001 |
| Low (n = 3,080) | 205 (66·6) | 537·1 (431·8–668·2) | <0.001 | 234 (76·0) | 245·1 (205·8–291·9) | <0.001 | 270 (87·7) | 70·02 (60·8–80·7) | <0.001 |
| **Non-Hispanic black** | | | | | | | | | |
| Normal (n = 923,131) | 127 (0·1) | 1 (ref) | | 411 (0·4) | 1 (ref) | | 2,518 (2·7) | 1 (ref) | |
| Intermediate (n = 11,816) | 80 (6·8) | 41·4 (31·07–55·3) | <0.001 | 123 (10·4) | 20·2 (16·4–24·9) | <0.001 | 199 (16·8) | 5·8 (5·03–6·8) | <0.001 |
| Low (n = 3,931) | 172 (43·8) | 286·7 (225·6–364·3) | <0.001 | 212 (53·9) | 114·2 (95·7–136·1) | <0.001 | 245 (62·3) | 23·3 (20·3–26·8) | <0.001 |
| **Non-Hispanic Asian** | | | | | | | | | |
| Normal (n = 447,842) | 51 (0·1) | 1 (ref) | | 98 (0·2) | 1 (ref) | | 352 (0·8) | 1 (ref) | |
| Intermediate (n = 2,919) | 16 (5·5) | 45·8 (25·6–82·02) | <0.001 | 22 (7·5) | 32·5 (20·2–52·5) | <0.001 | 38 (13·0) | 16·3 (11·5–23·0) | <0.001 |
| Low (n = 785) | 47 (59·9) | 545·8 (354·6–840·07) | <0.001 | 56 (71·3) | 335·4 (234·1–480·4) | <0.001 | 60 (76·4) | 100·4 (74·5–135·4) | <0.001 |
| **Non-Hispanic Other** | | | | | | | | | |
| Normal (n = 209,253) | 25 (0·1) | 1 (ref) | | 88 (0·4) | 1 (ref) | | 572 (2·7) | 1 (ref) | |
| Intermediate (n = 2,464) | 9 (3·7) | 27·0 (12·4–58·5) | <0.001 | 16 (6·5) | 14·0 (8·1–24·1) | <0.001 | 29 (11·8) | 4·1 (2·8–6·0) | <0.001 |
| Low (n = 698) | 41 (58·7) | 414·1 (242·6–707·0) | <0.001 | 46 (65·9) | 133·02 (89·9–196·9) | <0.001 | 55 (78·8) | 26·8 (19·8–36·3) | <0.001 |

*Wald p-value

AOR, adjusted odds ratios; BMI, body mass index; CI, associated 95% confidence intervals.

Odds ratios and 95% CIs were adjusted infant sex, maternal age, maternal smoking status, infant birth weight, maternal education, maternal BMI, previous number of live births, and gestational age.

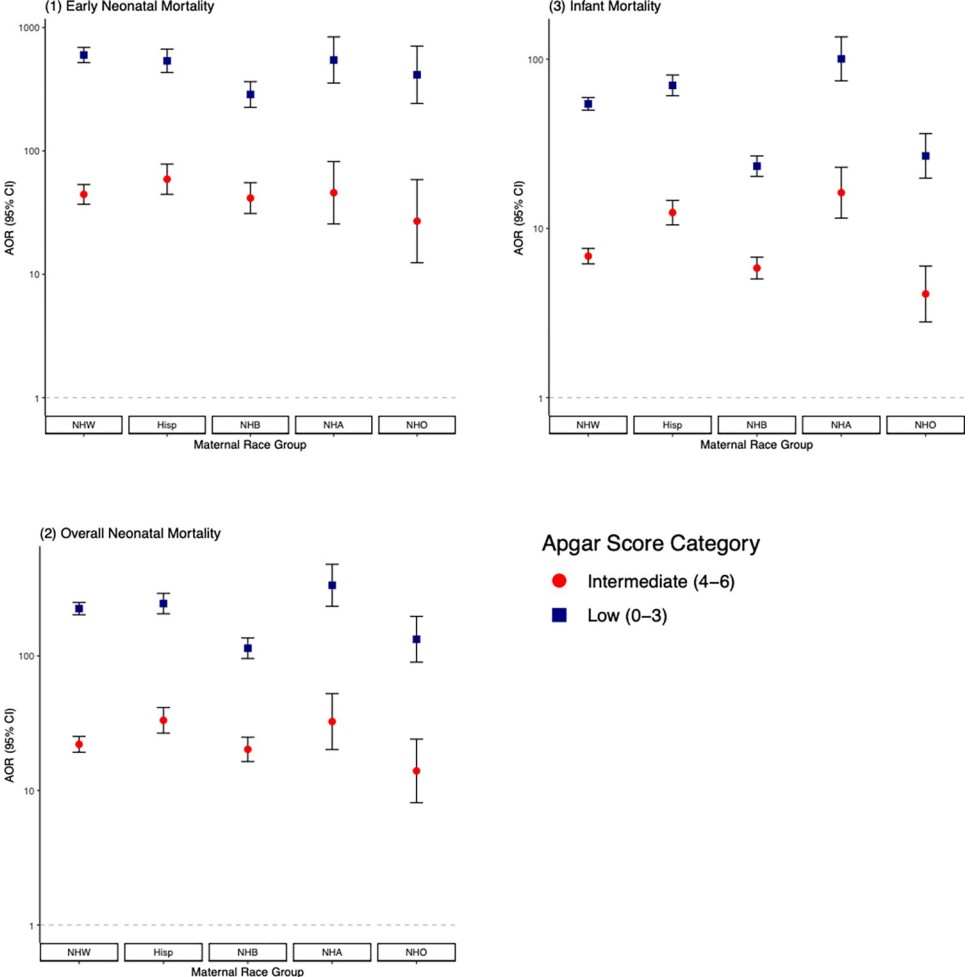

**Fig 2. AORs and 95% CIs for early neonatal, overall neonatal, and infant mortality in relation to Apgar score, stratified by maternal race group (*n* = 6,809,653).** AORs of (1) early neonatal mortality; (2) overall neonatal mortality; and (3) infant mortality for infants with low (0 to 3) and intermediate (4 to 6) 5-minute Apgar scores referent to infants with normal (7 to 10) 5-minute Apgar scores. ORs were adjusted for infant sex, maternal age, maternal smoking status, infant birth weight, maternal education, maternal BMI, previous number of live births and gestational age. AOR, adjusted odds ratio; CI, confidence interval; Hisp, Hispanic; NHA, non-Hispanic Asian; NHB, non-Hispanic black, NHO, non-Hispanic other; NHW, non-Hispanic white.

births [24,30]. The difference in rates is likely due to restricting the study population to term infants born without congenital malformations.

We report that the 5-minute Apgar score has a strong association with infant mortality in a large multiracial population in a real-world setting. Therefore, the score can be used for informing prognosis and as a valid metric in epidemiological studies. However, there are differences in the strength of association between Apgar score and mortality between racial groups, which should be taken in to account in clinical practice and research studies.

Reduced strength of association between the 5-minute Apgar score and mortality in black and non-Hispanic non-Asian groups might be explained in part by systematic differences in assignment of score at birth. The potential differential assignment of the score by race therefore requires further investigation; we suggest that particular consideration be given to individual components of the score, particularly the "skin color" component. The "skin color" scheme relies on classifying infants as blue, pale, or pink and is unlikely to be equally

efficacious across a range of skin tones, as demonstrated by a recent study stating that a majority of physicians do not agree that "pink all over" is an accurate description of vigorous African-American infants [8,31]. It is possible that refinement of the scoring system to capture circulatory status more reliably could improve its performance in identifying infants at high risk of mortality. However, the reasons for the attenuated association between Apgar score and mortality in black and Hispanic infants are certainly more complex than just differences in how the score is assigned at birth and are likely to be driven by the social drivers behind poor outcomes in certain race groups, which are not necessarily captured by clinical scores such as the Apgar. Further research is needed to understand the complex social and structural pathways that explain the differential associations between Apgar scores and mortality across racial groups in order to inform future use of the score.

This study has a number of strengths. The large sample size derived from a population of all births in the US from 2016 to 2017 allowed for a study cohort that was representative of the US population and minimized selection bias. All of the data were derived from routinely collected NCHS data, minimizing recall and social desirability biases. The present study included 5 maternal racial groups in analysis, allowing for a more thorough analysis of racial differences than previous studies.

The study also has limitations, largely due to the nature of routinely collected data. There were missing values for some covariates, and due to the low frequency of missing values and lack of significant associations between them, these missing values were included as "unknown" categories. The analysis considered maternal race as the infant's race due to a high proportion of missing data for paternal race, which will have results in some misclassification for infants where paternal race was different than maternal race. This analysis excluded preterm births due to concerns about the reliability of the Apgar score in this population; however, a recent study has demonstrated a strong association between low Apgar score and neonatal mortality in preterm infants [3,20]. Further analysis should be conducted to assess how the association between Apgar score and mortality varies by race group among preterm infants.

A further limitation may have been that models were not adjusted for all maternal comorbidities or medication use, because the data to do so reliably were not available.

## Conclusions

Overall, low 5-minute Apgar scores are strongly associated with early neonatal, overall neonatal, and infant mortality outcomes in a large multiracial population. There are strong associations between race and the 5-minute Apgar score, with black infants having higher odds of being assigned a low score than their white counterparts. Strong associations also exist between race and mortality within the first year of life, with black and non-Hispanic other infants having the highest odds of neonatal and infant mortality. Importantly, there are racial differences in the strength of the association between Apgar score and mortality. This suggests that while Apgar scores should continue to be used in clinical and research settings, practitioners and researchers should be aware that both the assignment and predictive ability of the Apgar score varies across racial groups.

## Supporting information

**S1 Table. STROBE checklist.**
(DOCX)

**S2 Table. Maternal race recategorizations.**
(DOCX)

**S3 Table. Mortality rates per 1,000 births, stratified by Apgar score and maternal race.**
(DOCX)

**S4 Table. Unadjusted odds ratios for early neonatal, overall neonatal, and infant mortality by race group.**
(DOCX)

**S5 Table. Unadjusted and adjusted odds ratios for mortality for multivariable models in non-Hispanic white cohort.**
(DOCX)

**S6 Table. Unadjusted and adjusted odds ratios for mortality for multivariable models in Hispanic cohort.**
(DOCX)

**S7 Table. Unadjusted and adjusted odds ratios for mortality for multivariable models in non-Hispanic black cohort.**
(DOCX)

**S8 Table. Unadjusted and adjusted odds ratios for mortality for multivariable models in non-Hispanic Asian cohort.**
(DOCX)

**S9 Table. Unadjusted and adjusted odds ratios for mortality for multivariable models in non-Hispanic other cohort.**
(DOCX)

## Author Contributions

**Conceptualization:** Emma Gillette, Sarah J. Stock.

**Data curation:** Emma Gillette, Clara Calvert, Sarah J. Stock.

**Formal analysis:** Emma Gillette, Clara Calvert.

**Investigation:** James P. Boardman, Clara Calvert, Sarah J. Stock.

**Methodology:** Emma Gillette, James P. Boardman, Sarah J. Stock.

**Project administration:** Emma Gillette, Sarah J. Stock.

**Resources:** Emma Gillette.

**Software:** Emma Gillette.

**Supervision:** James P. Boardman, Sarah J. Stock.

**Validation:** James P. Boardman, Clara Calvert, Jeeva John.

**Writing – original draft:** Emma Gillette.

**Writing – review & editing:** Emma Gillette, James P. Boardman, Clara Calvert, Jeeva John, Sarah J. Stock.

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
