## [Editor Report · Decision Letter 0]

10 Dec 2021

Dear Dr Stock, 

Thank you for submitting your manuscript entitled "Racial Differences in the Association of Low Apgar Scores and Mortality in the United States: a cohort study of 6,809,653 infants" for consideration by PLOS Medicine.

Your manuscript has now been evaluated by the PLOS Medicine editorial staff and I am writing to let you know that we would like to send your submission out for external peer review.

Please re-submit your manuscript within two working days, i.e. by Dec 14 2021 11:59PM.

Kind regards,

Beryne Odeny

PLOS Medicine

---

## [Decision Letter · Decision Letter 1]

10 Mar 2022

Dear Dr. Stock,

Thank you very much for submitting your manuscript "Racial Differences in the Association of Low Apgar Scores and Mortality in the United States: a cohort study of 6,809,653 infants" (PMEDICINE-D-21-05026R1) for consideration at PLOS Medicine. 

Your paper was evaluated by a senior editor and discussed among all the editors here. It was also sent to independent reviewers, including a statistical reviewer. The reviews are appended at the bottom of this email and any accompanying reviewer attachments can be seen via the link below:

Considering these reviews, we would be grateful if you could please revise your manuscript to respond to comments raised by reviewers. We would strongly recommend that you pay special attention to the reviewer #2's comments regarding the premise of your research question and your approach. Please note that further consideration is dependent on the submission of a manuscript that addresses all reviewer concerns. 

[LINK]

In light of these reviews, I am afraid that we will not be able to accept the manuscript for publication in the journal in its current form, but we would like to consider a revised version that addresses the reviewers' and editors' comments. Obviously we cannot make any decision about publication until we have seen the revised manuscript and your response, and we plan to seek re-review by one or more of the reviewers. 

We expect to receive your revised manuscript by Mar 31 2022 11:59PM. Please email us (plosmedicine@plos.org) if you have any questions or concerns.

We look forward to receiving your revised manuscript. 

Sincerely,

Beryne Odeny, 

PLOS Medicine

plosmedicine.org

1) Please revise your title according to PLOS Medicine's style. Your title must be nondeclarative and not a question. It should begin with main concept if possible. Please place the study design in the subtitle (i.e., after a colon), e.g., a retrospective cohort study. 

2) Please include line numbers in your next draft.

3) Is there a chance you can obtain more recent data from this setting?

4) Please conclude the “Introduction” with a clear description of the study question or hypothesis. The description of the database can be moved to the methods section.

5) Abstract:

a) Please ensure that all numbers presented in the abstract are present and identical to numbers presented in the main manuscript text.

b) Please quantify the main results (please present both 95% CIs and p values).

c) In the last sentence of the Abstract Methods and Findings section, please describe the main limitation(s) of the study's methodology.

7) Did your study have a prospective protocol or analysis plan? Please state this (either way) early in the Methods section. 

8) Thank you for providing the STROBE checklist as Supporting Information. 

a) Please add the following statement, or similar, to the Methods: "This study is reported as per the Strengthening the Reporting of Observational Studies in Epidemiology (STROBE) guideline (S1 Checklist)."

9) In your statistical analyses, please account for clustering of observations at hospital and county levels. Generalized Estimating Equations (GEE) or hierarchical/ multilevel models, among others, may be useful in this case.

10) Did you adjust for maternal comorbidity or medication use? If not, please consider this or acknowledge the lack of adjustment as a limitation 

11) Please provide p values in addition to 95% CIs in the main text and tables

12) Please define the abbreviations in Tables and Figures e.g, GED, BMI, SD

13) Please remove the “Role of the funding source”, “Data sharing”, and “conflict of interest” statement in the methods section. This information is captured in the metadata obtained in the submission form

14) Please remove the “Competing Interests”, “Data availability,” and “Funding” statements at the end of the main text. This information is captured in the metadata obtained in the submission form.

15) References: 

a) Please select the PLOS Medicine reference style in your citation manager. In-text reference call outs should be presented as follows noting the absence of spaces within the square brackets, e.g., "... services [1,2]."

b) References should have no more and no less than six names before et al. For references with more than six names, please ensure that et al., is inserted after six names

c) Please ensure that journal name abbreviations consistently match those found in the National Center for Biotechnology Information (NCBI) databases. https://journals.plos.org/plosmedicine/s/submission-guidelines#loc-references. 

Comments from the reviewers:

Reviewer #1: Thank you for the opportunity to review this manuscript. This is overall a well written paper. 

1. My comments are mainly related to the test of interactions. By using stratified analysis, the author claims that the association between Apgar scores and mortality varies across racial groups. However, whether the difference in the association between Apgar scores and mortality across racial groups is statistically significant or not has not been tested or presented. In order to acheive this, the author may need to used the adjusted model with interaction beteeen Apgar Score category and racial group. With one racial group as a reference group, ratio of AOR needs to be tested and presented.

2. The resoluiton of Figure 2 needs to be increased.

Reviewer #2: This study examines assess whether the association between low Apgar score and mortality in infants varies across racial groups.

This is a well written paper, and the analysis appears to be appropriate (having said this I would recommend that the paper is reviewed by a statistical reviewer). 

The sample size is of course impressive - but I am afraid that this cannot remedy the other concerns I have with the paper.

A large part of the introduction is making the argument that while Apgar scores have been widely used and are highly predictive of infant mortality, the scores have largely been developed and validated in White populations. I suspect the authors are speaking about predictive validity - mortality. If so, this is of course an important issue. The authors also talk about validating the scores to assess infant status at birth. This is of course an entirely different matter and would involve a study comparing the gold standard judgement (presumably conducted by a physician) with the scores recorded on a birth chart/card. The authors even describe how Black infants have been shown to be assigned lower Apgar scores than White infants. This was reinforced by the later description of how there was a lack of research on the 'racial differences in the applicability of the scores..". 

Reading this, I was inclined to conclude that validating Apgar scores (not just predictive validity) or trying to understand these differentials might be the focus of the study. But this is not the case. The paper, in my reading simply shows that in a country where Black infants are twice as likely to die as White infants (cited by the authors) that Apgar scores predict this and that Non-Hispanic Black and Non-Hispanic Other groups have higher odds of being assigned a low score. Without wishing to sound dismissive - how is this interesting? Is that not exactly what would be expected. 

Had the paper been set up to show how systematic racism was implicated in the assigning of Apgar scores - i.e. higher or lower scores compared to gold standard and compared to other race groups - now that would be really interesting. If the study was able to show how Apgar scores differed by race from gold standard (irrespective of infant birth status) - that would be interesting.

As a result I do not think the paper passes muster in terms of novelty for a public health journal of the stature of PLOS Medicine. It should definitely be published but perhaps in a specialist journal such as an Obstetric journal. 

Reviewer #3: Thank you for the opportunity to review this manuscript. In this paper, the authors examine the association of low Apgar scores and infant mortality across different racial groups using vital statistics data of infants born in the US. The paper finds that the five-minute Apgar score is strongly associated with infant mortality but identified differences in the strength of association between across different racial groups. Specifically, the strength of association between low Apgar score and mortality was reduced in Black and non-Hispanic non- Asian groups suggestive of reduced predictive ability of mortality in these groups.

The authors have examined an important topic that could make a significant contribution to the literature. The study has some significant implications for the interpretation and applicability of Apgar scores and their prediction of mortality across different racial groups. 

Overall, this is a good paper but could be improved by being more in-depth in its descriptions and analysis of the findings and the meaning of the results as currently some sections are quite superficial. The methods section could be more detailed and the results section is currently very brief. The authors need to more specifically outline what the findings mean for the application and use of Apgar scores across different racial groups apart from stating that caution should be taken when interpreting the scores. I was expecting to see a section on why was the study done, what do these findings mean etc. to really highlight the key takeaway points - this would be helpful in drawing out the key findings of this paper as well. 

More attention should be paid to detail as there were repeating text in the manuscript and parts that have not been updated after previous revisions (e.g STROBE statement page numbers)

I have made some further specific comments below for the authors consideration to revise the manuscript.

Specific comments 

ABSTRACT

p2, Methods and findings: The detailed description in the results of how many infants of each race were included and excluded cases is not essential for the abstract. Perhaps you can keep the % and drop the n. I would suggest to review this and focus on presenting the most important results.

INTRODUCTION

p3, para 1-2: I would recommend the text in paragraph 2 of the introduction be moved up into paragraph 1 (after the first sentence) and the detail about the Apgar score from the second sentence be moved down into a separate paragraph. This is so the problem your study is investigating is upfront. The detail of the components of Apgar score is not so important that it needs to be in the first paragraph.

p4, para 2: missing word - "and" before "discrimination"

p5, para 1: The last sentence of the introduction ("The database consists of…") is not essential here and would be more appropriate in the methods section where a description of the data source is provided.

METHODS 

The ethics sub-section can be placed towards the end of the methods section. The ethics statement is also repeated in the Study Population section at the beginning of page 4. This can be removed.

The methods section does not follow the STROBE checklist and the page numbers in the checklist do not match the current manuscript (e.g there is no Study Design section in the methods but the checklist in the Appendix indicates it is on pages 8-11 in the manuscript). This should be the first section or combined with study population. I would suggest the authors structure the methods section using similar headings to the strobe statement. The "Database" sub-heading should be "Data sources" for example.

Also, the current STROBE statement checklist is missing the first column that indicates which section of the manuscript the information can be found.

p7, para 2: Outcomes, exposures, covariates - this section needs to be better organised. I would suggest describing the outcome variable first followed by the explanatory variables.

p8, top paragraph: There is a bit of a mix up or description of analysis and variables in this section as well - the part on what was adjusted for should be in the analysis section but the definition of the variables should remain in this section. Handling of missing values also belongs in the analysis section.

Figure 1: please improve the quality of the flow diagram (font is currently blurred) and consider reducing amount of text and perhaps include the use of boxes.

RESULTS

p10, para 1: All of the numbers and % in the table do not need to be repeated in the text. Suggest to mention the % for race without the numbers in the text.

On the other hand, on p13 there is no description of the magnitude of the odds described at all. Instead of stating that there is only a lower or higher odds, the authors can be more specific - for example, "…non-Hispanic black infants had almost two times higher odds of being assigned a low Apgar score…" Otherwise the information in the text is not very informative.

p15-16. Please indicate which table the statement made in the third sentence in paragraph 1 is located.

There is no need to describe the chi-square test here - this belongs in the methods/analysis section.

The main results/tables of your study are in the Appendix - I would suggest to move (some) of these to the main manuscript. The unadjusted analyses I would think were less important and could be in the appendix. 

Overall, the description of your results is very brief given the amount of results and tables included.

Tables S6-S8 in the Appendix need to be fit to the page as the last columns are cut off. Suggest to include the word documents with the original tables rather than an image.

DISCUSSION

p16, first para: The first paragraph of a discussion should highlight the most important findngs of the paper. However the authors describe the unadjusted results. 

Please also avoid repeating results but rather explaining or interpreting the data in the discussion 

Reviewer #4: Thanks for allowing me to review your very interesting manuscript. I have several comments, and some questions which may lead to some revisions.

1. I am curious as to why you excluded births to mothers under the age of 15? Whilst the numbers are likely to be small they are a high-risk group, more likely to have births of compromised babies due to fetal growth restriction and other problems of placental dysfunction. I note that you did include women who did not complete 8th grade at school and if this group is included then perhaps the very young should also be there. Could you elaborate on your reasons for their exclusion?

2. Similarly, the inclusion of births up to 44 + 6 weeks of gestation is unusual as this might imply a problem of data accuracy. Even without intervention, which is almost universal now before 42 + 6 weeks, very few pregnancies would progress beyond 43 weeks and certainly not beyond 44. Why such a high figure?

3. What is the origin of the list of congenital abnormalities that have been excluded? It contains many obvious severe conditions (e.g., anencephaly) but also some others which are quite common and not really associated with infant mortality (e.g., hypospadias). 

4. I noted that the explanation about the exemption from full ethics review is repeated in the sections on Ethics and Study Population which is a duplication.

5.The overall smoking rates are much lower than I would have expected (6.8%). In my own country (Australia) we are still seeing rates of 10-15% overall, and these vary a lot between different ethnic groups. Can you comment on this, and if you can validate this figure against any other data source? As it is a risk factor for both neonatal and infant mortality it is important to get this right in any analysis, especially when calculation the adjusted OR.

6. Finally, I am curious as to the predictive value of the Apgar scores for both neonatal mortality (the original use) and later infant mortality. Whilst some causes of infant mortality are linked to conditions which have their origins in the perinatal period many others are not necessarily linked (e.g. trauma, drowning etc). Have you looked separately at the infant deaths which occur beyond 28 days (about 50% of the total cohort of deaths), rather than including all deaths in the infant mortality numbers, which includes the early and late neonatal ones

Reviewer #5: Thank you for the opportunity to review this manuscript. Overall the analyses are well conducted and the manuscript well written. I have a few minor edits to suggest:

1. The following text should be combined combined with the Inclusion and Exclusion Criteria to make a single continuous paragraph on study population, which also means that the gestational age criteria do not need to be stated twice: 

"Study Population

This population cohort study evaluated singleton infants born between 37+0 and 44+6 weeks between January 1, 2016 and December 31, 2017 in the United States." 

2. The 2nd sentence of the Study Population paragraph should be added to the Database paragraph: "All data were nonidentifiable and publicly available through the NCHS Division of Vital Statistics and complied with the NCHS, Centers for Disease Control and Prevention (CDC) Data User 6

Agreement Terms and Conditions.23,24"

3. The rest of the Study Population paragraph is an exact repeat from the Ethics paragraph and should be deleted, except the last sentence on Strobe reporting guidelines which can be moved to the end of the methods section.

4. I did not quite follow the logic for the following: "the findings suggest that the odds of mortality based on a low Apgar score are underestimated in these groups" and suggest this statement be deleted. The remainder of the discussion and conclusion are much clearer in their interpretations.

[LINK]

---

## [Decision Letter · Decision Letter 2]

10 May 2022

Dear Dr. Stock,

Thank you very much for re-submitting your manuscript "Associations between low Apgar scores and mortality by race in the United States: a cohort study of 6,809,653 infants" (PMEDICINE-D-21-05026R2) for review by PLOS Medicine.

I have discussed the paper with my colleagues and it was also seen again by three reviewers. I am pleased to say that provided the remaining editorial and production issues are dealt with we are planning to accept the paper for publication in the journal.

[LINK]

We look forward to receiving the revised manuscript by May 17 2022 11:59PM.   

Sincerely,

Beryne Odeny, 

PLOS Medicine

plosmedicine.org

Requests from Editors:

1) Abstract:

a) Please structure your abstract using the PLOS Medicine headings (Background, Methods and Findings, Conclusions).

b) Please combine the Methods and Findings sections into one section, “Methods and findings”

c) In the last sentence of the Abstract Methods and Findings section, please describe the main limitation(s) of the study's methodology

Comments from Reviewers:

Reviewer #1: Thank you for addressing reviewers' comments and improving the manuscript. My questions in the previous round has been suffuciently answered.

1. In Table 2 and Table 3, why there's only one p-value for multiple tests for interaction? For instance, in Table 2, there should be seperate p-values for Non-Hispanic white vs Hispanic, and Non-Hispanic white vs Non-hispanic black. In table 3, there should be p-value for normal vs intermedaite comparison, and p-value for normal vs low comparison.

2. According to the Office of Management and Budget Standars for the Classification of Federal Data on Race and Ethnicity, Hispanic ethnicity is recorded as distinct from race. In this paper, the author used different category and treated Hispanic as a mutually exclusive racial group from other racial groups. Explanation is needed for the selction of the categories for the analysis.

Reviewer #2: The authors have responded to reviewer comments

Reviewer #3: Thank you to the authors for their efforts to address the comments and revise the manuscript. The revisions made have improved the manuscript somewhat but I'm afraid there remain some substantial issues which need to be addressed particularly in relation to the methods section and discussion. I have outlined these below:

Minor revisions

ABSTRACT

Line 45 - Stating the limitations is not needed in the abstract

Line 49 (findings) - In the sentence, "A total of 6, 728,829 infants…" the percentages should be included after the numbers for easier interpretation. It's not clear why they were removed from the previous version.

Line 52 - abbreviations (AOR) are usually not used in an abstract - please check with journal requirements. As it is not used multiple times in the abstract it may not be necessary.

AUTHORS' SUMMARY

This is a very nice overall summary. Just one small suggestion:

Line 90 - please remove the "n" as it doesn't add anything here and it is stated clearly in the title of the paper.

Major revisions

METHODS

The methods section needs reorganisation - grouping study design, setting and participants is not following the STROBE guidelines and these are all very different topics, especially participants (study setting may not be needed). This section now also discusses the exclusion and inclusion criteria. 

There are now two sections with the heading of "variables" one called "quantitative variables" - all Generally, there should be description of the variables - dependent and independent. The statistical analysis should be separate. Please closely review the STROBE guidelines and other similar papers for order and content of the methods section.

I do not have statistical expertise but is this multivariable regression not multivariate?

Line 213 (Statistical methods and Quantitative Variables) - Please separate out this section into paragraphs as it is too long as one piece of text.

Line 251 - this statement on STROBE should be at the beginning of the methods not in the Ethics statement.

RESULTS

All of the figures are still very low quality and not clear. 

Table 1 - this table is very hard to read and has no lines - please format and add some spacing between the variables. 

DISCUSSION

There seems to be a disconnect between the first paragraph of the discussion and the author's summary around the implications of the findings. The discussion is very positive and supportive of the Apgar score while the Authors' summary is more critical indicating that Apgar scores are less useful among non-White populations. The overall conclusions should be based on the results only and need to be consistently communicated throughout the paper. 

In my previous review I indicated that the authors should avoid only restating the results in the discussion but instead elaborate on what their results mean. Instead of doing this the text was just removed. The discussion cannot consist only of a summary and implications. The results need to be discussed more adequately and critically.

[LINK]

---

## [Decision Letter · Decision Letter 3]

31 May 2022

Dear Dr. Stock,

Thank you very much for re-submitting your manuscript "Associations between low Apgar scores and mortality by race in the United States: a cohort study of 6,809,653 infants" (PMEDICINE-D-21-05026R3) for review by PLOS Medicine.

I have looked at the paper and it was also seen again by one reviewer. I am pleased to say that provided the remaining minor editorial and production issues are dealt with we are planning to accept the paper for publication in the journal.

[LINK]

We look forward to receiving the revised manuscript by Jun 07 2022 11:59PM.   

Sincerely,

Beryne Odeny, 

Senior Editor 

PLOS Medicine

plosmedicine.org

Requests from Editors:

1. Abstract: please revise the subheading "Interpretation" to "Conclusion" in line with PLOS Medicine's style

2. Discussion section: please delete the subheadings including "Findings in context", "Implications" and "Strengths and Limitations"

Comments from Reviewers:

Reviewer #3: Thank you to the authors for the revisions to the manuscript. The paper is now much clearer and flows well. I have no further comments on this version. 

I look forward to seeing this published.

[LINK]

---

## [Editor Report · Decision Letter 4]

1 Jun 2022

Dear Dr Stock, 

On behalf of my colleagues and the Academic Editor, Dr. Mark Tomlinson, I am pleased to inform you that we have agreed to publish your manuscript "Associations between low Apgar scores and mortality by race in the United States: a cohort study of 6,809,653 infants" (PMEDICINE-D-21-05026R4) in PLOS Medicine.

PRESS

Sincerely, 

Beryne Odeny 

PLOS Medicine